# Effects of Red and Fermented Ginseng and Ginsenosides on Allergic Disorders

**DOI:** 10.3390/biom10040634

**Published:** 2020-04-20

**Authors:** Myung Joo Han, Dong-Hyun Kim

**Affiliations:** 1Department of Food and Nutrition, Kyung Hee University, Seoul 02447, Korea; mjhan@khu.ac.kr; 2Neurobiota Research Center, Department of Pharmacy, Kyung Hee University, Seoul 02447, Korea

**Keywords:** *Panax* sp., ginsenosides, polysaccharides, allergy, immune system

## Abstract

Both white ginseng (WG, dried root of *Panax* sp.) and red ginseng (RG, steamed and dried root of *Panax* sp.) are reported to exhibit a variety of pharmacological effects such as anticancer, antidiabetic, and neuroprotective activities. These ginsengs contain hydrophilic sugar-conjugated ginsenosides and polysaccharides as the bioactive constituents. When taken orally, their hydrophilic constituents are metabolized into hydrophobic ginsenosides compound K, Rh1, and Rh2 that are absorbable into the blood. These metabolites exhibit the pharmacological effects more strongly than hydrophilic parental constituents. To enforce these metabolites, fermented WG and RG are developed. Moreover, natural products including ginseng are frequently used for the treatment of allergic disorders. Therefore, this review introduces the current knowledge related to the effectiveness of ginseng on allergic disorders including asthma, allergic rhinitis, atopic dermatitis, and pruritus. We discuss how ginseng, its constituents, and its metabolites regulate allergy-related immune responses. We also describe how ginseng controls allergic disorders.

## 1. Introduction

Allergies including asthma, allergic rhinitis, atopic dermatitis (AD), atopic conjunctivitis, and anaphylaxis are common, persistent, and incorrigible disorders [1,2]. The prevalence of allergies ranges from 10% to 40% of the population worldwide [2]. A variety of drugs, including immune modulators and biological agents, have been developed for the treatment of allergies [3,4]. However, they have certain limitations due to their side effects: glucocorticoids often induce the adrenal insufficiency and cause infections and skin atrophy; calcineurin inhibitors cause neurotoxicity, nephrotoxicity, infections, and skin cancers; and biological agents such as omalizumab increase infections and tumor development [5,6]. Therefore, natural products with fewer adverse effects such as red ginseng and radix glycyrrhizae have been frequently used as the functional foods and traditional Chinese medicines [7,8]. Many studies have been conducted on their anti-allergic effects. Of these, we focused on the anti-allergic effects of ginseng and the constituent ginsenosides and polysaccharides in the present review.

## 2. Chemistry of Ginseng

The term ginseng is used to represent the dried root of the *Panax* spp. (family Araliaceae), including *Panax ginseng* Meyer (Korean ginseng), *Panax quiquifolium* L. (American Ginseng), and *Panax notoginseng* (Burk.) FHChen (notoginseng) [9,10]. When the fresh roots of these *Panax* spp., particularly Korean ginseng, are dried or steamed/dried, they are named white ginseng (WG) or red ginseng (RG), respectively. Ginseng has been used world-wide as herbal medicine or functional food for promoting vitality, increasing the resistance to stress, and modulating immune responses [11,12]. The bioactive constituents are considered to be ginsenosides, such as protopanxadiol-type, protopanaxatriol-type, and oleanane-type ginsenosides, and polysaccharides such as ginsan [13] (Figure 1). Of protopanaxadiol-type ginsenosides, ginsenosides Rb1, Rb2, Rc, Rd, and Rg3 and quinquenosides I and II are highly isolated from ginseng [14,15]. Of protopanaxatriol-type ginsenosides, ginsenosides Rg1 and Re are frequently isolated [16,17]. Of oleanane-type ginsenosides, ginsenoside Ro and chikusetsusaponin are isolated [10,18,19].

## 3. The Role of Gut-Microbiota-Mediated Metabolism in the Mediation of Biological Effects of Ginseng

Korean ginseng, American ginseng, and notoginseng all contain hydrophilic sugar-conjugated ginsenosides and polysaccharides as the bioactive components [13]. Of ginsenosides, hydrophilic ginsenosides Rb1, Rb2, Rc, and Re have a variety of pharmacological activities such as anti-inflammatory, antidiabetic, hepatoprotective, and anticancer activities in the in vivo studies [13,20]. However, when these ginsenosides or ginseng extracts are orally gavaged, these ginsenosides such as Rb1 and Re are not easily absorbed into the blood [21,22] (Figure 2). Therefore, these contact with gut microbiota, which transform hydrophobic metabolites such as compound K (CK) and ginsenoside Rh1. These metabolites such as CK are detected in the blood rather than parental constituents [21,22]. In addition, when ginsenoside Rb1 was orally administered in germ-free rats, Rb1 and CK both were not detected in the blood [23]. To understand the reason for this, when ginsenosides were incubated with fecal bacteria, they were strongly and quickly transformed into CK [24]. Orally administrated ginsenoside Rb1, a main constituent of *Panax ginseng*, is transformed to CK through ginsenosides Rd and F2 in humans and animals by gut bacteria, such as *Bifidobacterium* sp. and *Bacteroides* sp., and thereafter these metabolites are detected in the blood and urine [25,26,27]. The absorption of gut-microbiota-mediated metabolites from ginseng constituent ginsenosides are significantly affected by intestinal environmental factors such as diets and antibiotics [28,29,30,31]. The biological activities of ginsenosides Rb1 and Re, such as anti-inflammatory and anti-allergic activities, are attenuated in mice by oral gavage of antibiotics [30,31]. When RG extract is orally administered in humans and mice, ginsenoside Rd is the most highly detected, followed by ginsenoside Rg3, ginsenoside Rg1, and protopanaxatriol [32,33]. However, when bifidobacteria-fermented red ginseng (fRG) is orally administered, ginsenoside Rd is the most highly detected, followed by ginsenoside Rg1, CK, ginsenoside Rg3, and protopanaxtriol [32,33]. The contents of these ginsenosides except ginsenoside Rg3 absorbed into the blood are significantly higher in the fRG-treated volunteers and mice than in the RG-treated volunteers and mice. However, when notoginseng extract, whose which main constituents are ginsenosides Ra3, Rb1, Rd, Re, and Rg1 and notoginsenoside R1, is orally administered to rats, the compounds mainly absorbed to the blood are ginsenosides Ra2, Rb1, and Rd, including CK [34]. This is controversial. Nevertheless, of parental ginsenosides and their metabolites, CK, ginsenosides Rh1, Rh2 and protopanaxatriol, which are hydrophobic metabolites of ginsenosides by gut microbiota, exhibit the most potent biological effects compared to those of parental compounds [31,32,33,35,36,37,38,39,40]. These results suggest that when ginseng extracts are orally administered, their hydrophilic constituents are metabolized by gut microbiota and their metabolites absorbable into the blood can express pharmacological effects: the pharmacological activities of ginseng extracts may be dependent on the absorbable metabolites produced by gut microbiota.

## 4. Anti-Allergic Effects of Ginseng

Ginseng extracts including RG extracts have been used in the traditional Chinese medicine for the treatment of allergic diseases including asthma, rhinitis, and AD [7,8,41]. Actually, many studies have been performed to support their anti-allergic effects in vitro, in animals, and in patients with allergic disorders.

### 4.1. The In Vitro and In Vivo Anti-Allergic Effects of Ginseng

The anti-allergic effects of ginseng have been mainly studied in vitro, in animals, and in patients with allergic disorders (Table 1). First, Kim and Yang evaluated the effects of WG on ovalbumin-induced asthma in mice [42]. They found that intraperitoneally injected RG restored the ovalbumin-induced expression of eosinophil major basic protein (EMBP), interleukin (IL)-1β, IL-4, IL-5, and tumor necrosis factor (TNF)-α expression in lung tissues. RG inhibited the ovalbumin-induced numbers of goblet cells and mitogen-activated protein kinases (MAPKs) in the bronchoalveolar lavage fluid of mice. Babayigit et al. reported that orally administered RG extract suppressed the chronic airway inflammation and mast cell populations in ovalbumin-sensitized mice [43]. Oral administration of WG or RG alleviated IL-4, IL-5, and IL-13 expression and immune cell infiltration in the bronchoalveolar regions of mice with ovalbumin-induced asthma [44]. They also suppressed IgE levels. Of these, RG more strongly lowered IgE level. Lee et al. reported that RG and fRG reduced serum IgE and ovalbumin-specific IgE levels and intestinal mucosal mast cell protease (MMCP)-1, IL-4, TNF-α, cyclooxygenase (COX)-2, and inducible NO synthase (iNOS) expression in ovalbumin-sensitized mice [45]. Furthermore, RG and fRG inhibited IL-4 expression in phorbol 12-myristate-13-acetate/A23187-stimulated RBL-2H3 cells and alleviated ovalbumin-induced allergic rhinitis in mice [46]. In particular, fRG potently reduced nasal allergy symptoms; IgE level in the blood; IL-4 and IL-5 levels in nasal mucosa; and mast cell, eosinophil, and Th2 cell populations in bronchoalveolar lavage fluid and restored ovalbumin-induced gut dysbiosis. The inhibitory effects of fRG in the treatment of allergic rhinitis were better than those of RG. Jung et al. reported that RG suppressed IL-4 and IL-5 levels and eosinophil populations in the nasal lavage fluid of ovalbumin-sensitized mice [47]. RG increased ovalbumin-suppressed splenic IL-12 expression; IFN-γ-to-IL-4 ratio; and small intestinal CD8-, IFNγ-, and IgA-positive cell populations in ovalbumin-sensitized mice [48]. Furthermore, fRG treatment improved the activities and emotions of quality of life. These results suggest that RG and fRG can alleviate allergic rhinitis in mice by suppressing IgE, IL-4, IL-5, and IL-13 expression and restoring altered gut microbiota and that fRG may display anti-allergic rhinitis activity more strongly than RG did due to the richness of absorbable ginsenosides.

In addition, Lee and Cho reported that RG suppressed mast cell populations, pruritic sensation, and IL-31 expression in NC/Nga mice with 2, 4, 6-trinitro-1-chrolobenzene (TNCB)-induced AD [49]. They also found that RG extract suppressed the ear thickness, IgE levels in the blood, and regulatory (FOXP3^+^) T cell and Langerhans cell (CD1a^+^) populations in the lesions of TNCB-sensitized NC/Ng mice [50]. Treatment with RG inhibited thymic stromal lymphopoietin (TSLP) and TNF-α expression and Langerhans cell populations in NC/Nga mice with TNCB-induced AD [51]. Kim et al. reported that topical application of RG significantly suppressed the clinical skin severity score in NC/Nga mice with TNCB-induced AD [52]. Furthermore, RG treatment decreased the mast cell infiltration and TNF-α and IL-4 expression in the TNCB-exposed lesions but did not affect IgE levels in the blood. Sohn et al. reported that RG decreased IgE levels in the blood and IL-4 and IL-10 expression, MAPKs activity, and NF-κB-independent Ikaros activation in the dorsal surface of mice with 1-chloro-2, 4-dinitrobenzene (DNCB)-induced AD [53]. RG decreased the IL-6, thymic stromal lymphopoietin (TSLP), and TNF-α, and thymus and activation-regulated chemokine (TARC) expression; MAPKs activation; and dermatitis score in DNCB sensitized mice [54]. The topical pretreatment with RG prevented the induction of ear swelling, nerve growth factor expression, and nerve fiber extension in mice by exposure to 2, 4-dinitrofluorobenzene (DNFB) [55]. RG treatment suppressed mammalian target of rapamycin (mTOR)/p70 ribosomal protein S6 kinase (p70S6K) signaling in anti-FcεRIa antibody-stimulated human basophil KH812 cells and DNFB-sensitized mice [56]. Choi et al. reported that cultivated Korean ginseng (CG) inhibited TNF-α/IFN-γ-induced thymus and activation-regulated chemokine (TARC) expression through NF-κB-dependent signaling in HaCaT cells [64]. Furthermore, CG ameliorated DNCB-induced atopic dermatitis severity; IgE and TARC expression in the blood; and TARC, TNF-α, IFN-γ, IL-4, IL-5, and IL-13 expression in the skin lesions of mice. Bae et al. reported that RG suppressed oxazolone-induced ear skin edema, IL-1β, TNF-α, and COX-2 expression in mice and inhibited iNOS and COX-2 expression in lipopolysaccharide-induced RAW264.7 cells [57]. Kang et al. reported that γ-irradiated black ginseng extract reduced the IgE/antigen-complex-induced degranulation in RBL-2H3 mast cells and alleviated the AD-like skin symptoms, IgE and IL-4 expression, and leukocyte populations in the blood [65]. These findings suggest that ginseng including RG and CG can suppress allergen-induced IgE level, TNF-α, TSLP, IL-4, and IL-6 expression, resulting in the attenuation of AD.

Lee et al. reported that RG strongly inhibited chloroquine-induced scratching in mice [58]. Furthermore, RG inhibited chloroquine-induced Ca^2+^ influx in the primary culture of mouse dorsal root ganglia. RG also showed an anti-pruritic effect in mice with histamine-induced scratching by blocking the histamine-induced histamine receptor type 1/TRPV1 pathway in sensory neurons [59]. Trinh et al. reported that RG extract inhibited IgE/antigen-induced passive cutaneous anaphylaxis reaction in mice and inhibited the IgE/antigen-stimulated degranulation and IL-4 expression in basophils [60]. They also found that ginseng and RG extracts inhibited compound 48/80-induced scratching behaviors, IL-4 expression, and NF-κB activation in mice [61]. Hwang et al. reported that the fermentation of ginseng with *Lactobacillus plantarum* inhibited the IgE-DNP-stimulated IL-4 expression in RBH-2H3 mast cells and passive cutaneous anaphylaxis in mice [63]. Park and Park evaluated the effects of RG on the regeneration of the full-thickness skin wounds in rat [66]. They also found that oral or topical treatments with RG significantly suppressed the wound size and accelerated tissue regeneration rate. RG significantly increased the gene expression levels of transforming growth factor-β1 and vascular endothelial growth factor during the early stages of wound healing. RG treatment increased matrix metalloproteinase (MMP)-1 and MMP-9 expression. Kim et al. reported that RG alleviated epidermal growth factor (EGF)-induced damage by blocking NF-κB and ERK in NCI-H292 cells and EGF-stimulated human airway epithelial cells [67]. These results suggest that RG and fRG can alleviate anaphylaxis and pruritus by suppressing IgE level, IL-4 and IL-5 expression, and NF-κB activation.

Based on these findings, ginseng including RG and fRG alleviates the acute and chronic phases of allergic diseases by modulating the innate and adaptive immune cells. Thus, ginseng can suppress IgE, IL-4, and IL-5 expression and Th2-to-Th1 cell ratio through the modulation of mast cell, basophil, and eosinophil activation, resulting in the attenuation of AD, allergic rhinitis, asthma, and pruritus. The anti-allergic effects of WG and RG were enforced by the fermentation. Their constituents may affect several pathways involved in allergic diseases by specific and nonspecific action mechanisms. However, their anti-allergic effects can be influenced by the quality and quantity of anti-allergic constituents found in the ginseng and the administered route. Therefore, the preparation of ginseng products must be standardized and well-characterized.

### 4.2. Efficacy of Ginseng in Patients with Allergic Disorders

A few reports are available on the clinical effectiveness of ginsengs against allergic disorders (Table 2). Kim et al. reported that in an 8-week, open, noncomparative clinical study of patients with AD, RG decreased eczema area and severity index score, transepidermal water loss, visual analogue scale, and sleep disturbance [68]. Jung et al. reported that in an open, noncomparative clinical study of patients with allergic rhinitis, RG alleviated rhinorrhea, nasal itching, and eye itching and suppressed IgE, IL-4 levels and eosinophil counts [69]. Park et al. reported that in a randomized, double-blind, placebo-controlled trial of patients with cold hypersensitivity in the hands and feet (CHHF), RG increased skin temperature of the hands and feet and decreased visual analog scale score of CHHF severity [70]. Jung et al. reported that in a 4-week, double-blind, placebo-controlled study of patients with persistent perennial allergic rhinitis, there was no significant difference in the total nasal symptom score between the fRG-treated and placebo groups in the experimental period, while the fRG-treated group, but not placebo group, showed the alleviation of nasal congestion [71]. These results suggest that RG and fRG may alleviate AD, allergic rhinitis, and cold hypersensitivity. Although several clinical trials have demonstrated effects of ginsengs in patients with allergic disorders, further controlled studies are required to clearly elucidate these effects.

## 5. Anti-Allergic Effects of Ginseng Constituents

In order to search for bioactive constituents of ginseng to treat allergic disorders, many researchers have examined the anti-allergic effectiveness of their constituent ginsenosides and polysaccharides in in vitro and in vivo studies [7,72,73] (Table 3). Of the ginsenosides, Rb1 inhibited IL-4 and GATA3 expression, airway resistance, and eosinophil population in the bronchoalveolar lavage fluid of ovalbumin-sensitized mice, while interferon-γ (IFNγ) and T-bet expression were increased [74]. Rb1 also inhibited compound 48/80-induced scratching behaviors in mice, while the IgE/complex-induced degranulation and IL-4 expression were not affected in RBL-2H3 cells [60,61]. Ginsenoside Rd suppressed ovalbumin-induced expression of IgE, IL-4, IL-5, and IL-13 in nasal mucosa and bronchoalveolar lavage fluid and alleviated gut dysbiosis in mice, resulting in the attenuation of allergic rhinitis [46]. Ginsenoside Rd enhanced Th1-response to *Candida albicans* surface mannan extract in mice [75]. Wang et al. reported that, of tested ginsenosides Rb1, Rd, F2, CK, and 20(S)-protopanaxadiol, ginsenoside F2 most potently inhibited the compound 48/80-stimulated degranulation of mast cells and RBL-2H3 cells [76]. Oh et al. found that ginsenoside Rg1 significantly reduced ovalbumin-induced increases in TSLP, IL-1β, and IL-4 expression; histamine and IgE levels; and eosinophil and mast cell populations in mice, while interferon-γ expression was enhanced [77]. Ginsenoside Rg1 inhibited NF-κB signaling pathways in cultured mast cells in vitro [77]. The combination of ginsenoside Rg1 with aluminum hydroxide strongly induced immune responses to ovalbumin in mice [78]. Lee et al. reported that ginsenoside Rg3 inhibited chloroquine-induced Ca^2+^ influx in primary culture of mouse dorsal root ganglia [58]. Furthermore, ginsenoside Rg3 significantly reduced chloroquine-induced scratching in mice. Lee et al. reported that ginsenoside Rg3 inhibited NF-κB activation and COX-2 expression in IL-1β-stimulated human asthmatic airway epithelial tissues [79]. Ginsenoside Rh2 attenuates allergic airway inflammation in ovalbumin-sensitized mice by regulating NF-κB activation and p38 MAPK phosphorylation [80]. RG saponin fraction and ginsenosides Rg3 and Rh2 inhibited compound 48/80- or histamine-induced scratching behavior and vascular permeability [61]. Ginsenosides Rg3 and Rh2 inhibited IL-4 and TNF-α expression in IgE/antigen-complex-stimulated RBL-2H3 cells [61,81].

Bae et al. examined the inhibitory effects of ginsenosides Rg3, Rf, and Rh2 on IgE/antigen-complex-induced passive cutaneous anaphylaxis in mice [82]. Of these, ginsenoside Rh2 most potently inhibited the IgE/antigen-complex-induced passive cutaneous anaphylaxis reaction. Ginsenoside Rh2 strongly inhibited the IgE/antigen-complex-induced RBL-2H3 cell degranulation [82]. Ginsenoside Rh2 also inhibited oxazolone-induced expression of COX-2, IL-1β, and TNF-α in the ears of mice, while the IL-4 expression was not affected [83]. Kim et al. reported that topical application of ginsenosides Rh2 or Rh2 plus Rg3 significantly reduced the clinical skin severity scores, ear thickness, mast cell populations, and TNF-α and IL-4 expression in the skin lesions of mice with TNCB-sensitized AD [52], while IFNγ expression and IgE levels were not affected.

Oral administration of ginsenoside Rh1 reduced AD-like clinical symptoms, ear swelling, IL-4, and IgE levels in the skin lesions of hairless mice with oxazolone-induced AD, while IFNγ and Foxp3 expression were increased [38]. Ginsenoside Rh1 also inhibited the release of histamine from rat peritoneal mast cells and the IgE/antigen-complex-induced passive cutaneous anaphylaxis reaction in mice. Ginsenoside Rh1 increased the membrane-stabilizing action in mast cells and inhibited COX-2 expression and NF-κB activation in RAW 264.7 cells. Park et al. reported that CK, a metabolite of ginsenoside Rb1, inhibited NO and prostaglandin E2 production in lipopolysaccharide-induced RAW 264.7 cells more strongly than the parental ginsenoside Rb1 [84]. CK also reduced the COX-2 expression and NF-κB activation. CK inhibited the IgE/antigen-complex-induced cell degranulation in RBL-2H3 cells and oxazolone-induced chronic dermatitis in mice [36]. Lin et al. reported that CK improved the accelerated and severe lupus nephritis in mice by blunting NLRP3 inflammasome activation and regulating T cell functions [85]. CK and its derivatives inhibited IgE production in mice with ovalbumin-sensitized asthma [86]. Shin et al. examined the anti-pruritic and vascular-permeability-inhibitory effects of ginsenoside Rb1 and its metabolite CK in mice with compound 48/80-, substance P-, or histamine-induced scratching behaviors [87]. When orally administered, ginsenoside Rb1 and CK both suppressed pruritic behaviors and skin vascular permeability. However, the intraperitoneal injection of ginsenoside Rb1 did not inhibit compound 48/80-induced scratching behaviors, while CK potently inhibited scratching behavior. Moreover, CK-fortified ginseng extract alleviated *Dermatophagoides farinae* body extract induced dermatitis score, ear thickness, scratching time, severity of skin lesions, and eosinophil and mast cell populations in NC/Nga mice [88]. These results suggest that ginsenosides and their metabolites can alleviate asthma, allergic rhinitis, AD, and scratching behavior by inhibiting IgE and IL-4 expression, NF-κB activation, and Ca^2+^ influx; increasing IFNγ expression; and stabilizing the degranulation of mast cells and basophils. Of ginsenosides and their metabolites, the most absorbable ginsenosides Rh1 and CK the most potently can alleviate AD, allergic rhinitis, pruritus, and anaphylaxis in vivo and in vitro, followed by Rd.

Ginseng polysaccharides isolated from *Panax japonicus* or *Panax ginseng*, the immunity-potentiating anti-cancer agents, stimulated immune response; thus, they activate phagocytosis, natural killer cell activity, and cytotoxic T cell activity [89]. Furthermore, they activate the phagocytosis of neutrophils and macrophages [90]. Ginsan isolated from *Panax ginseng* reduced ovalbumin-sensitized IL-5 expression and airway hyperresponsiveness, remodeling, and eosinophilia in mice, resulting in the attenuation of asthma [91]. RG-II isolated from *Panax ginseng* induced the Th1/Th2 immune response and IFNγ expression in mice with ovalbumin-induced asthma, while IL-4 and GATA3 expression and eosinophil populations were decreased in the bronchoalveolar lavage fluid [92]. CVT-E002 derived from North American Ginseng also activated Th1 responses and increased IL-10 expression, resulting in the attenuation of allergic airway inflammation and airway hyperresponsiveness [93]. These suggest that ginseng polysaccharides can stimulate the Th1 cell immune responses, resulting in the attenuation of asthma with the suppression of Th2 cell activation.

In addition, Lee et al. reported that oral intake of Korean ginseng could induce anaphylaxis in occupational settings by non-IgE-dependently activating basophil/mast cells [94]. Hon and Leung et al. reported that urticarial could occur in a feeding neonate, whose mother took American ginseng [95]. Erdle et al. reported that a child experienced anaphylaxis after inhaling powered American ginseng [96]. These results suggest that ginseng must be carefully used in clinic, because it can cause side effects due to its allergic reactions.

## 6. Gut Microbiota Enforce Anti-Allergic Activities of Ginseng Constituents

Ginseng extracts and their constituents, particularly ginsenosides, showed anti-allergic effects in the in vivo studies. However, the absorption of these ginsenosides into the blood is not easy due to their hydrophilicity. Therefore, they are metabolized by gut microbiota in the intestine, which transform hydrophilic ginsenosides such as ginsenosides Rb1, Rb2, and Re into hydrophobic ginsenosides such as CK and Rh1 [13,20,97,98]. Comparing the anti-allergic activities of naïve ginsenosides Rb1, Rg3, and Re to those of their metabolite ginsenosides CK, Rh2, and Rh1, metabolites (ginsenosides CK, Rh2, and Rh1) suppress allergic reactions such as passive cutaneous anaphylaxis, scratching, and asthma more potently than parental ginsenosides Rb1, Rg3, and Re, respectively [38,62,81,82,93]. However, oral gavage of antibacterials suppresses their biotransformations and attenuates anti-allergic activities in mice. For example, when antibiotics (cefadroxil, oxytetracycline, and erythromycin mixture; COE), are orally gavaged, the fecal ginsenoside Re-metabolizing β-glucosidase and α-rhamnosidase activities and production of the metabolite ginsenoside Rh1 production are significantly suppressed [98]. The metabolism of ginsenoside Rb1 by gut microbiota is also inhibited by antibiotic treatment [31,97,98]. Furthermore, the anti-allergic activity of orally gavaged ginsenoside Re is significantly attenuated in mice treated with COE, but that of orally gavaged ginsenoside Rh1 are not affected. Oral gavage of ginsenoside Rh1 inhibits IL-4 and TNF-α expression and NF-κB and c-jun activation in mice with histamine-stimulated scratching more potently than parental ginsenoside Re. These results suggest that orally administered ginseng extracts and their hydrophilic ginsenosides should be metabolized to hydrophobic ginsenosides by gut microbiota, which enhances their anti-allergic activity, and, when simultaneously treated with antibacterials, their anti-allergic activities are attenuated. In addition, ginseng extract has been reported to rarely activate allergic responses rather than attenuate allergic disorders [95,96,97]. However, the mechanism should be clarified to safely use ginseng for the treatment of allergic disorders in clinics.

## 7. Conclusions

Herein, we discussed the current knowledge related to the effectiveness of ginseng on allergic disorders including asthma, allergic rhinitis, AD, and pruritus. Many studies are limited in examining the effectiveness of ginseng, including red ginseng and fRG and their constituent ginsenosides Rb1, Rd, and Rg3, against allergic disorders. Nevertheless, ginseng extracts alleviate allergic disorders such as asthma, allergic rhinitis, AD, and pruritus by inhibiting IgE, IL-4, and IL-5 expression through the modulation of mast cells, eosinophils, and Th1-to-Th2 ratio (Figure 3). Of ginseng extracts, fRG most potently alleviates allergic disorders, followed by RG and WG. Of their ginsenosides, CK, Rh1, and Rh2, which are the metabolites from hydrophilic parental ginsenosides by gut microbiota, strongly alleviated allergic disorders. To enforce these metabolites, fRG was developed. These ginseng constituents, absorbable into the blood, should express pharmacological effects, and the pharmacological activities of ginseng extracts may be dependent on the absorbable metabolites produced by gut microbiota. Ginseng itself can be allergen. Moreover, the great part of these results are inconclusive in the quality and quantity of anti-allergic constituents found in the ginseng and the administered route. Therefore, the anti-allergic effects of ginseng must be supported by further clinical and in-depth in vitro and in vivo studies.

## Figures and Tables

**Figure 1 biomolecules-10-00634-f001:**
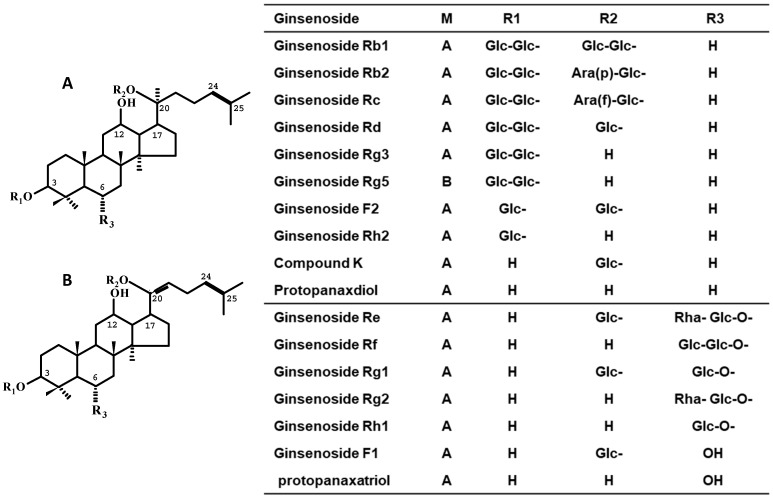
The Representative Ginsenosides Contained in WG and RG.

**Figure 2 biomolecules-10-00634-f002:**
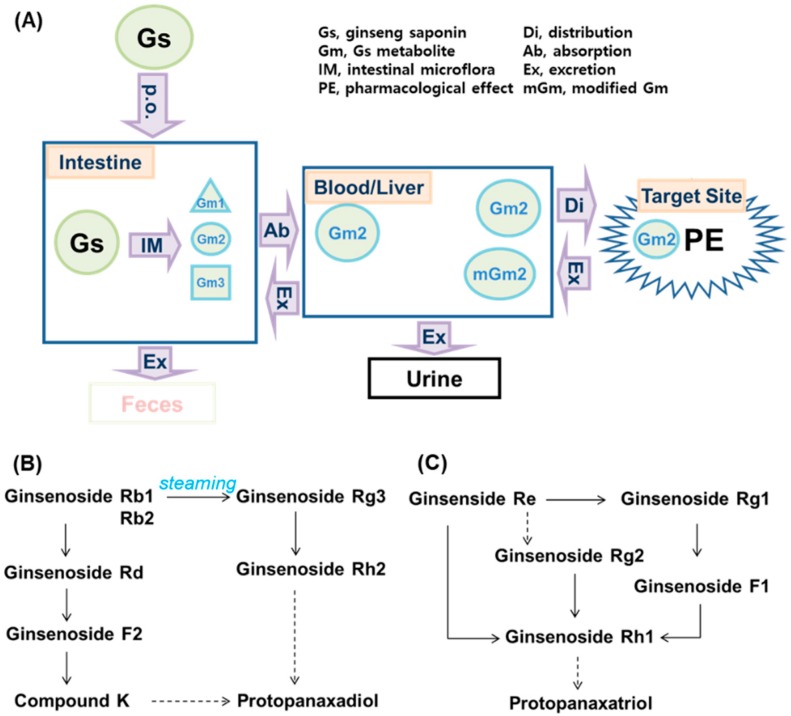
The proposed metabolic pathway of ginseng and its constituent ginsenosides by gut microbiota. (**A**) The fate of orally administered ginseng saponins in vivo. (**B**) The metabolic pathway of protopanaxadiol-type ginsenosides. (**C**) The metabolic pathway of protopanaxatriol-type ginsenosides. Solid arrows, potently proceeded; dashed arrows, weakly proceeded.

**Figure 3 biomolecules-10-00634-f003:**
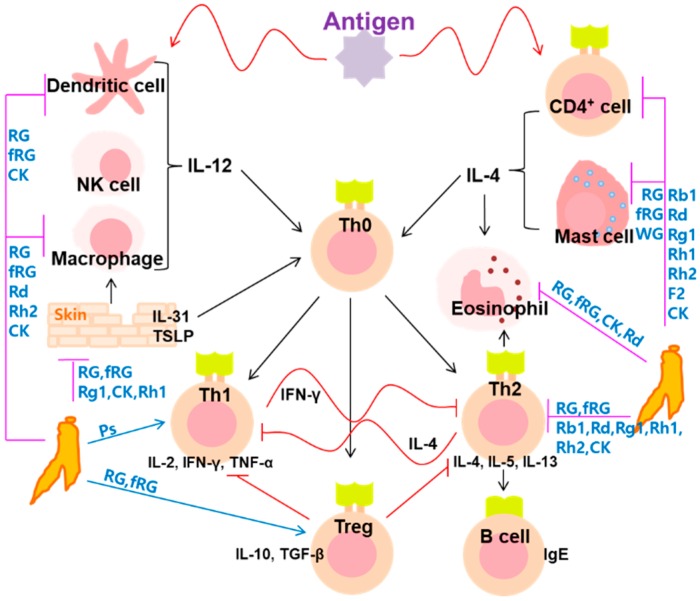
The hypothetical antiallergic action mechanisms of ginseng and ginsenosides. CK, compound K; fRG, fermented red ginseng; RG, red ginseng; WG, white ginseng.

**Table 1 biomolecules-10-00634-t001:** Summary of ginseng extract effects on allergic disorders.

Ginseng	Effect	Dosage/Ad. Route	Ref
RG	*in mice with OVA-sensitized asthma/rhinitis* Suppressed IgE, MMCP-1, IL-1β, IL-4, IL-5, IL-13, TNF-α, COX-2, and iNOS expression; MMPK and NF-κB activation; and mast cell, eosinophil, and Th2 cell populations Increased ovalbumin-suppressed splenic IL-12 expression; IFN-γ-to-IL-4 ratio; and small intestinal CD8-, IFNγ-, and IgA-positive cell populations Alleviated chronic airway inflammation, nasal allergy symptoms, and gut microbiota dysbiosis	B, 2 g/kg, *p.o. 7 d* B, 30 m/kg, *p.o.* 7 d B, 0.2%, in diet 8 w B, 50 mg/kg, *p.o.* 6 d B, 2 g/kg, *p.o*. 14 d B, 60 mg/kg, *p.o*. 37 d	[43] [44] [45] [46] [47] [48]
	*in NC/Nga mice with TNCB-induced AD* Suppressed IgE, IL-4, IL-10, IL-31, TNF-α, and TSLP expression; MAPKs activity; and NF-κB-independent Ikaros activation. Suppressed mast cell, Treg cell, and Langerhans cell populations. Suppressed ear thickness, clinical skin severity, and pruritic sensation	200 mg/kg, *p.o.* 10 d 200 mg/kg, *p.o*. 3 w 200 mg/kg, *p.o.* 5 d 0.1%, s.a. 21 d	[49] [50] [51] [52]
	*in mice with DNCB/DNFB-induced AD* Decreased IgE, IL-4,IL-6, IL-10, TSLP, TNF-α, nerve growth factor, and TARC expression and MAPK and p70S6K signaling Decreased ear swelling and dermatitis score	B, 0.1%, *s.a*. 2 w B, 400 mg/kg, *p.o*. 6 w B, 1%, *s.a*. 8 d B, 1%, *s.a*. 8 d	[53] [54] [55] [56]
	*in mice with oxazolone-induced AD* Inhibited IL-1β, TNF-α, and COX-2 expression Suppressed ear skin edema	I, 0.1%, *s.a.* 7 d	[57]
	*in mice with pruritus* Inhibited chloroquine-induced scratching, histamine-induced scratching, and compound 48/80-induced scratching behaviors Inhibited histamine receptor type 1/TRPV1 pathway and Ca^2+^ influx	I, 100 mg/m, *p.o.* I, 50 mg/m, *p.o.* I, 200 mg/kg, *p.o.* B, 200 mg/kg, *p.o.*	[58] [59] [60] [61]
	*in mice with PCA reaction* Inhibited IgE/antigen-induced PCA reaction Inhibited the IgE/antigen-stimulated degranulation, IL-4 expression, NF-κB activation in basophils	B, 200 mg/kg, *p.o.* I, 50 mg/kg, *p.o.*	[61] [62]
WG	*in mice with OVA-sensitized asthma/rhinitis* Suppressed EMBP, IL-1β, IL-4, and IL-5 expression and MMPK activity Suppressed IL-4, IL-5, and IL-13 expression and immune cell infiltration	B, 20 mg/kg, *i.p. 3 d* B, 30 m/kg, *p.o.* 7 d	[42] [44]
fRG	*in mice with OVA-sensitized asthma/rhinitis* Inhibited nasal allergy symptoms and gut dysbiosis Suppressed IgE, IL-4, and IL-5 expression and mast cell, eosinophil, Th2 cell, Th2/Th1 populations	B, 0.2%, in diet, 8 w B, 50 mg/kg, *p.o.* 6 d B, 60 mg/kg, *p.o*. 37 d	[45] [46] [48]
	*in mice with PCA reaction* Inhibited IgE-DNP-stimulated passive cutaneous anaphylaxis in mice Inhibited IgE-DNP-stimulated IL-4 expression in RBH-2H3 mast cells	B, 50 mg/kg, *p.o.*	[63]
CG	*in NC/Nga mice with DNCB-induced AD* Inhibited IgE; TNF-α/IFN-γ-induced TARC, TNF-α, IFN-γ, IL-4, IL-5; and IL-13 expression Ameliorated dermatitis severity	20 mg/kg, *s.a*. 4 w	[64]
BG	*in mice with DNCB-induced AD* Reduced IgE and IL-4 expression and leukocyte populations Alleviated the AD-like skin symptoms	B, 100 mg/kg, *s.a.* 4 w	[65]

AD, atopic dermatitis; Ad, administered; B, Balb/c; BG, γ-irradiated black ginseng; CG, cultivated ginseng; COX, cyclooxygenase; d, day; DNCB, 1-chloro-2,4-dinitrobenzene; DNFB, 2,4-dinitrofluorobenzene; EMBP, eosinophil major basic protein; fRG, fermented red ginseng; I, ICR; IFN, interferon; iNOS, inducible NO synthase; m, mouse; MAPK, mitogen-activated protein kinase; MMCP, mucosal mast cell protease; OVA, ovalbumin; PCA, passive cutaneous anaphylaxis; p.o., per oral; s.a., skin application; TARC, thymus and activation-regulated chemokine; TNCB, 2,4,6-trinitro-1-chrolobenzene; TNF, tumor necrosis factor; Treg, regulatory T; TSLP, thymic stromal lymphopoietin; w, week; WG, white ginseng.

**Table 2 biomolecules-10-00634-t002:** Summary of ginseng extract effects on allergic disorders.

Ginseng	Effect	Ref
RG	*in an 8-week, open, noncomparative clinical study of patients with AD (1–2 g/day)* decreased eczema area and severity index score, transepidermal water loss, visual analogue scale decreased sleep disturbance	[68]
	*in an open, noncomparative clinical study of patients with allergic rhinitis (3 mg/kg/day, 4 weeks)* alleviated rhinorrhea, nasal itching, and eye itching suppressed IgE, IL-4 levels and eosinophil counts	[69]
	*in a randomized, double-blind, placebo-controlled trial of patients with cold hypersensitivity in the hands and feet (CHHF) (1 g/day, 8 weeks)* increased skin temperature of the hands and feet decreased visual analog scale score of CHHF severity	[70]
fRG	*in double-blind, placebo-controlled study of patients with persistent perennial allergic rhinitis (750 mg/day, 4 weeks)* Alleviated nasal congestion and the activities and emotions of quality of life.	[71]

AD, atopic dermatitis; fRG, fermented red ginseng; RG, red giseng.

**Table 3 biomolecules-10-00634-t003:** Summary of ginseng constituent effects on allergic disorders.

Constituent	Effect	Ref
Ginsenoside Rb1	Suppressed IL-4 and GATA3 expression, airway resistance, and eosinophil cell population and increased IFNγ and T-bet expression in ovalbumin-sensitized mice Inhibited compound 48/80-induced scratching behaviors in mice Inhibited NO and prostaglandin E2 production in LPS-induced RAW 264.7 cells Inhibited IgE/antigen-induced degranulation of RBL-2H3 cells and PCA reaction in mice Inhibited compound 48/80-stimulated degranulation of mast cells and RBL-2H3 cells.	[36,60,62,74,84]
Ginsenoside Re	Suppressed histamine-induced IL-4 and TNF-α expression, NF-κB and c-jun activation, and scratching behaviors in mice	[38,97]
Ginsenoside Rd	Suppressed IgE, IL-4, IL-5, and IL-13 expression and allergic rhinitis and gut dysbiosis in ovalbumin-sensitized mice Enhanced Th1-response to *Candida albicans* surface mannan extract in mice Inhibited compound 48/80-stimulated degranulation of mast cells and RBL-2H3 cells	[46,75]
Ginsenoside Rg1	Reduced TSLP, IL-1β, and IL-4 expression; histamine and IgE secretion; and eosinophil and mast cell populations and increased IFNγ expression in ovalbumin-induced mice Inhibited NF-κB signaling pathways in cultured mast cells Induced immune responses to OVA in mice by the combination with aluminum hydroxide	[77,78]
Ginsenoside Rg3	Inhibited chloroquine-induced Ca^2+^ influx in primary culture of mouse dorsal root ganglia Reduced chloroquine-induced scratching in mice Inhibited NF-κB activation and COX-2 expression in IL-1β-induced human asthmatic airway epithelial tissues Alleviated allergic airway inflammation and suppressed NF-κB activation and p38 MAPK phosphorylation in OVA-sensitized mice Inhibited compound 48/80- or histamine-induced scratching behavior and vascular permeability Inhibited IL-4 and TNF-α expression in IgE/antigen-complex-stimulated RBL-2H3 cell Inhibited the IgE/antigen-complex-induced PCA reaction in mice Inhibited the IgE/antigen-complex-induced RBL-2H3 cell degranulation Reduced the clinical skin severity scores, ear thickness, mast cell populations, and TNF-α and IL-4 expression in the skin lesions of mice with TNCB-sensitized AD by the combination with Rh2	[52,58,79,82]
Ginsenoside Rh2	Suppressed allergic airway inflammation and suppressed NF-κB activation and p38 MAPK phosphorylation in OVA-sensitized mice Inhibited compound 48/80- or histamine-induced scratching behavior and vascular permeability Inhibited IL-4 and TNF-α expression in IgE/antigen-complex-stimulated RBL-2H3 cell Inhibited the IgE/antigen-complex-induced PCA reaction in mice Inhibited the IgE/antigen-complex-induced RBL-2H3 cell degranulation Inhibited oxazolone-induced expression of COX-2, IL-1β, and TNF-γ in the ears of mice Reduced the clinical skin severity scores, ear thickness, mast cell populations, and TNF-α and IL-4 expression in the skin lesions of mice with TNCB-sensitized AD	[52,61,80,81,82,83]
Ginsenoside Rh1	Reduced AD-like clinical symptoms, ear swellings, IL-4, and IgE expression and increased IFNγ and Foxp3 in mice with oxazolone-induced AD Inhibited the release of histamine from rat peritoneal mast cells and the IgE/antigen-complex-induced PCA reaction in mice Increased the membrane-stabilizing action in mast cells Inhibited COX-2 expression and NF-κB activation in RAW 264.7 cells Inhibited histamine-induced IL-4 and TNF-α expression, NF-κB and c-jun activation, and scratching behaviors in mice	[38,97]
Compound K (CK)	Inhibited NO and prostaglandin E2 production, COX-2 expression, and NF-κB activation in LPS-induced RAW 264.7 cells Inhibited IgE/antigen-complex-induced cell degranulation in RBL-2H3 cells and oxazolone-induced chronic dermatitis in mice Improved the accelerated and severe lupus nephritis in mice Inhibited IgE production in mice with ovalbumin-sensitized asthma Inhibited compound 48/80-, substance P-, or histamine-induced scratching behaviors and vascular permeability in mice Inhibited IgE/antigen-induced degranulation of RBL-2H3 cells and PCA reaction in mice Inhibited compound 48/80-stimulated degranulation of mast cells and RBL-2H3 cells (CK-fortified ginseng extract) Alleviated *Dermatophagoides farinae* body extract induced dermatitis score, ear thickness, scratching time, severity of skin lesions, and eosinophil and mast cell populations in NC/Nga mice	[36,62,84,85,86,87,88]
Polysaccharide	*Ginsan* Reduced ovalbumin-sensitized IL-5 expression and airway hyperresponsiveness, remodeling, and eosinophilia (asthma) in mice *RG-II* Induced the Th1/Th2 immune response and IFNγ expression and suppressed IL-4 and GATA3 expression and eosinophil populations in mice with ovalbumin-induced asthma *CVT-E002* Activated Th1 responses, increased IL-10 expression, suppressed allergic airway inflammation and airway hyperresponsiveness	[91,92,93]

AD, atopic dermatitis; CK, compound K; COX, cyclooxygenase; IFN, interferon; LPS, lipopolysaccharide; MAPK, mitogen-activated protein kinase; OVA, ovalbumin; PCA, passive cutaneous anaphylaxis; TNCB, 2, 4, 6-trinitro-1-chrolobenzene; Th, helper T cell.

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
