# Peer review of "Effects of Red and Fermented Ginseng and Ginsenosides on Allergic Disorders"

_biomolecules, 2020, doi:10.3390/biom10040634_

Round 1

Reviewer 1 Report

This review contains good scientific data pertaining to ginseng on allergy.

Quality of this review could be improved by:

-better organization of the text, e.g. separating pre-clinical and clinical data

-better organization of pre-clinical data according on the biological pathway affected and route of administration and types of extracts (both text and table)

-Currently, there is a collection/presentation of data, but there is  lack of

   -critical evaluation of the data and study

   -synthesis of ideas, deduction, and conclusion.

There are good potentials, but more work is needed to address the deficiency.

Author Response

Quality of this review could be improved by:

-better organization of the text, e.g. separating pre-clinical and clinical data

-->Thank you for your comment. We revised it.

-better organization of pre-clinical data according on the biological pathway affected and route of administration and types of extracts (both text and table)

--> Thank you for your comment. We revised it.

-Currently, there is a collection/presentation of data, but there is lack of    -critical evaluation of the data and study    -synthesis of ideas, deduction, and conclusion.

--> Thank you for your comment. We revised it.

Reviewer 2 Report

Dear authors

Please see my comments here attached.

Best regards

Author Response

Effects of ginseng and ginsenosides on allergic disorders

General comments and questions.

The paper is well written (a moderate language checking is required since there are some mistakes all over the paper), logically organized and supported by recent references. Nonetheless, it needs more figures to explain how ginseng and its bioactive compounds exert their anti-allergic effects. Furthermore, the anti-allergic effects (sections 3 and 4) were not discussed. A brief discussion after each of the sections would improve the paper.

Chemical structures of the major bioactive compounds should be added.

A graphical abstract summarizing the anti-allergic effects of ginseng and ginsenosides should be added.

The section “3. How the Bioactive Constituents of Ginseng Express Their Biological Activities” could be summarized in a figure. It’d be of great interest for the readers.

The authors overused ALSO in their manuscript. Linking words’ use should be corrected.

The reported anti-allergic effects of ginseng were related to in vivo studies. It’d be better to add the in vitro studies, especially those related to the molecular mechanisms.

Perspectives should be added to improve the paper.

--> Thank you. We added 3 Figures and 1 Table according to your comment. We also added the discussion.

Specific comments

Line17. To enforce these metabolites, fermented WG and RG are developed

It’d be better to correct are ===== were Biomolecules Mar 2020

-->Thank you for your comment. We revised it.

Lines 19-20. Therefore, this review introduces the current knowledge on related to the effectiveness of ginseng on allergic disorders

On should be suppressed

--> Thank you for your comment. We revised it.

Lines 30-31. “….. glucocorticoids often induce the suppression of adrenal function”

It’d be correct to write . “….. glucocorticoids often induce adrenal insufficiency

-->Thank you for your comment. We revised it.

Lines 32-33. “…. biological agents increase infections and tumor development”

The biological agents causing infections and cancers should be defined since it is not correct to generalize.

--> Thank you for your comment. We revised it.

Lines 55-57. “However, when these ginsenosides or …………… the blood rather than parental constituents”

This sentence should be rewritten in more suitable and direct style. It should be cut into two sentences to be more understood.

--> Thank you for your comment. We revised it.

Lines 57-58. “In addition, when ginsenoside Rb1 was orally administered in germ-free rats, even if CK was not detected in the blood”

This sentence seems to be incomplete. It should be corrected.

-->Thank you for your comment. We revised it.

Line 75. It is controversy. Nevertheless,

It is controversy should be suppressed since nevertheless means the same.

--> Thank you for your comment. We revised it.

Line 82. “Allergic Effects of Ginseng”

The subtitle should be corrected to anti-allergic effects of Ginseng

-->Thank you for your comment. We revised it.

Line 157. “Table 1. Summary of ginseng extract effects on allergic disorders.”

It should be corrected to “Table 1. Summary of ginseng extract effects on allergic disorders in vivo.”

-->Thank you for your comment. We revised it.

Round 2

Reviewer 1 Report

The authors have made organizational changes as suggested.

Critical evaluation of data cited (including report of negative or inconsistent data) and synthesis of new ideas based on data presented is still missing. Conclusion at the end of each section was not adequate, it was merely a repeat of the data.

Other areas that required attention:

  1. Title should be changed from "ginseng" to "red and fermented ginseng" as this is the primary focus of this review.
  2. use of appropriate terminology re ginseng. eg Panax ginseng as Korean ginseng, South China ginseng as notoginseng. L189: Cultivated ginseng referred to what species?
  3. Chemistry. As this review dealt mostly with Red ginseng and fermented red ginseng, their unique chemistry should be included.
  4. Table 2. Should include information on ginseng treatment of animals. 
  5. Table 3 deals with effect of ginseng constituents, it should document treatment details.
  6. Section 3. title should be revised to role of metabolism in mediation of biological effect of ginseng.
  7. L151. Efficacies of RG were better than those of WG. No data was presented to support that! 
  8. Summary of data presented in Table 2 and 3 will be of interest to many readers, but it requires better overall interpretation of the composite data/information.
  9. Fig 3 was presented as part of conclusion of the review. It is not clear whether it was based on strong data or it was a hypothetical model that requires further research or testing?

Author Response

The authors have made organizational changes as suggested.

Critical evaluation of data cited (including report of negative or inconsistent data) and synthesis of new ideas based on data presented is still missing. Conclusion at the end of each section was not adequate, it was merely a repeat of the data.

-->Thank you. We revised our manuscript according to your comments.

Other areas that required attention:

1) Title should be changed from "ginseng" to "red and fermented ginseng" as this is the primary focus of this review.

-->Thank you. We revised it according to your comment

2) use of appropriate terminology re ginseng. eg Panax ginseng as Korean ginseng, South China ginseng as notoginseng. L189: Cultivated ginseng referred to what species?

-->Thank you. We revised it according to your comment.

3) Chemistry. As this review dealt mostly with Red ginseng and fermented red ginseng, their unique chemistry should be included.

-->Thank you. We revised it according to your comment.

4) Table 2. Should include information on ginseng treatment of animals. 

-->Thank you. We revised it according to your comment.

5) Table 3 deals with effect of ginseng constituents, it should document treatment details.

-->Thank you. We revised it according to your comment.

6)Section 3. title should be revised to role of metabolism in mediation of biological effect of ginseng.

-->Thank you. We revised it according to your comment.

7) L151. Efficacies of RG were better than those of WG. No data was presented to support that! 

-->Thank you. We revised it according to your comment.

8) Summary of data presented in Table 2 and 3 will be of interest to many readers, but it requires better overall interpretation of the composite data/information.

-->Thank you. We added the discussion according to your comment.

9) Fig 3 was presented as part of conclusion of the review. It is not clear whether it was based on strong data or it was a hypothetical model that requires further research or testing?

-->Thank you. We revised it according to your comment.